# Temporal and spatial mapping of theoretical biomass potential across the European Union

**Susann Günther**[1], **Tom Karras**[1], **Friederike Naegeli de Torres**[1], **Sebastian Semella**[1], **and Daniela Thrän**[1,2,3]

[1]Bioenergy Systems, Deutsches Biomasseforschungszentrum gemeinnützige GmbH, Leipzig, 04347, Germany
[2]Bioenergy, Helmholtz Centre for Environmental Research GmbH, Leipzig, 04318, Germany
[3]Faculty of Economic Science, University of Leipzig, 04103 Leipzig, Germany

**Correspondence:** Susann Günther (susann.guenther@dbfz.de)

**Abstract.** With the increasing challenge to shift our economic system from carbon to renewable energy carriers, the demand for biogenic resources is growing. Biogenic municipal waste, agricultural by-products and industrial residues are under-utilised but are increasingly gaining in value. To date, there is no continuous database for these resources in the EU-27 countries. Existing datasets that estimate resource potentials for a single point in time often lack validation. A reliable and continuous database is thus needed to support the growing bioeconomy.

Spatial and temporal high-resolution data of biogenic residues serve as an invaluable resource for identifying areas with significant theoretical biomass potential and allows an in-depth understanding of dynamic patterns over time. This study elucidates the theoretical biomass potentials of 13 distinct biomasses from municipal waste, agricultural by-products and industrial residues quantified annually from 2010–2020. The spatial scope of the research covers the EU-27 Member States incorporating all entities represented at various levels within the Nomenclature of Territorial Units for Statistics (NUTS) as delineated by Eurostat, where possible. The regionalised data are subsequently validated against regional statistics from different countries. The findings demonstrate the feasibility of creating a time series of theoretical biomass potentials for the 13 selected waste types, by-products, and residues, and underscore the critical role of data validation when regionalising national or sub-national data to smaller NUTS entities. It could be shown that the values of small regions (NUTS 3) correlated well on average. When looking at individual regions in detail, regional characteristics such as the location of cultivation, waste management or reporting methods could lead to over- or underestimates of up to 100 %. Therefore, data at the regional level provide only limited reliability. In the case of industrial residues, regionalisation gave good results localising preference regions of high theoretical biomass potential, but more data on industrial production are needed to also estimate residual quantities at sub-national and local levels.

The biomass potentials modelled in this study have been published in an open-access database, which is designed as an extensible tool, enabling the understanding of national and regional trends of theoretical biomass potentials in the European Union and of the reliability of the regionalised data.

The estimated theoretical potential dataset can be downloaded free of charge from: https://doi.org/10.48480/g53t-ks72 (Günther et al., 2023).

## 1 Introduction

The need to shift the economy from fossil to renewable resources is leading to a steadily rising demand for biogenic materials as a sustainable resource for energy and material use. With the introduction of the Circular Economy Action Plan as one pillar of the European Green Deal, the European Union (EU) demands that the rates of waste reuse and recycling be increased, as well as sustainable product design (European Commission, 2020). In order to achieve these ambitious goals, knowledge of available biogenic residues, by-products and wastes is essential, but data availability on the different resources varies greatly. Municipal solid waste (MSW) and its organic fraction is monitored by the EU statistic agency Eurostat on the national level. Data on MSW streams indicate that landfill declined from over 60 % of MSW treatment to 24 % over the last three decades. This was mainly achieved by increasing the rate of material recycling using composting and digestion of degradable wastes and incineration with an increase of each of the two waste treatment streams of over 10 % (eurostat, 2022). However, further efforts are needed to comply with the Waste Framework Directive (WFD), which explicitly demands a 65 % recycling rate of MSW by 2035 (European Commission, 2008). Countries such as Bulgaria, Croatia, Malta and Slovenia are still above 60 % landfill and will face challenges in the upcoming years to fulfil EU targets. The fraction of MSW composted or digestated increased slowly from 7 % to 18 % over the same period in the EU. Some countries such as Austria, Italy and the Netherlands have already managed to increase this rate to well above 20 %, while many other countries are below the EU average. These data clearly show the differences in the implementation of waste policies across EU countries and that a large share of the theoretical biomass potential of biowaste remains unused so far. However, sub-national data on waste generation or waste treatment are not collected by the EU and are, therefore, not provided by Eurostat. Similarly, there is no existing monitoring tool at the European level for agricultural by-products and industrial residues. Yet, in order to efficiently utilize biogenic residues, monitoring the local occurrence of waste and residual materials is of paramount importance.

Moreover, the review of Hamelin et al. (2019) on existing biomass potential studies in the EU reveals that the majority of studies estimate forest and agricultural by-products only at a national level. High-resolution estimates on NUTS 3 level are limited to the studies by Bellot et al. (2021), Hamelin et al. (2019) and Scarlat et al. (2019). The latter published rasterised datasets of 1 km pixel size, although Hamelin et al. (2019) and Scarlat et al. (2019) were published in peer review journals. The ENPRESSO database (Ruiz et al., 2019), which is an EU-wide dataset, addresses the biomass potentials of agricultural by-products and biogenic MSW but excludes industrial residues and is limited to NUTS 2 level. Only the biomass potential of biogenic municipal waste has also been addressed on the different NUTS levels in the EU project S2BIOM (Dees et al., 2017). These calculation models help us to understand the spatial distribution and, hence, the identification of areas with high biomass potentials of single resources. Nevertheless, all of these studies estimate the biomass potential for a single point in time only and in some cases try to estimate future potentials from a single reference year, while the implementation of political strategies and private investment need long-term planning and, hence, requires reliable time series of biomass potential development (Brosowski et al., 2019). This includes a solid data validation that provides information on the accuracy and reliability of a monitoring instrument.

It is apparent that the utilization of biogenic residues is rapidly gaining importance in the EU (Bell et al., 2018). This trend is not only reflected in the increase of fermentation of MSW, as shown in the Eurostat data above. A surge in funding and technological advances in this area underline the emerging interest for biogenic resources, especially for material uses. Numerous studies have highlighted the potential of deriving a variety of bulk chemicals from biogenic waste, residues or by-products which can, for example, serve as substitutes for petrochemicals and are, therefore, promising intermediates in a bioprocessing chemical industry (Iglesias et al., 2020; Di Domenico Ziero et al., 2020; Sheldon, 2014). However, the practicality and economic viability of transferring these concepts to large-scale industrial operations has yet to be conclusively demonstrated. For this reason, data on regional availability of suitable biogenic feedstocks are crucial in shaping the course of business case formulation and facilitating informed decision making.

In this study, a threefold approach is used to model biomass potentials. The theoretical biomass potentials of 12 residues from agriculture, municipal waste and industry are modelled on a yearly basis for the period of 2010–2020 and mapped for Europe with a spatial resolution of NUTS 0 to NUTS 3, where possible. The novelty of an 11-year time series which allows trend analysis of the theoretical biomass potential with a resolution up to NUTS 3 is especially crucial for volatile feedstocks such as agricultural by-products, which can be highly influenced by weather conditions. In addition, the available biomass potential can be influenced by policy frameworks in the form of waste regulations, using, e.g. MSW or quota regulations of agricultural production volumes. The spatial resolution of NUTS 3 additionally makes this time series interesting for investors and decision makers since the market of bio-based products is still developing and regulations are changing quickly (Siegfried et al., 2023).

The Nomenclature of Territorial Units for Statistics classification (NUTS) is a standard geocode reference developed and regulated by the EU and is available from Eurostat with four different spatial levels. Level NUTS 0 entities represent countries, level 1 entities represent major socio-economic regions, level 2 entities represent basic regions and level 3 en-

tities represent small regions. Regular revision of NUTS entities can lead to area and code changes, which have been taken into account in this study. All estimations in this paper and the dataset of Günther et al. (2023) refer to the official geocode reference from 2016.

The study, conducted in the scope of the Horizon 2020 BBI JU project "CAFIPLA" (GA. No.: 887115), considers the theoretical biomass potentials from agricultural by-products such as different straw types and sugar beet leaves, bio-waste from households, as well as industrial residues from sugar, beer and cheese production due to the suitability requirement for the designed pilot plant of the project. Straw and municipal solid waste are among the top 10 biomasses in Europe in terms of their technical potential, according to a literature review by Karras et al. (2022). Data from Eurostat, European industrial associations, commercial registers, as well as the CORINE Land Cover raster are used for the model. The theoretical potential is expressed in units of specific mass and in terms of fresh matter (FM) CE1. Finally, the modelled biomass potentials are validated against available statistical data, derived from various national statistic agencies to assess the quality and reliability of the regionalised data, not only to show the potential in comparison to other studies in this area but also to validate the method. Data on whey production is directly available from Eurostat (European Commission, 2021) and, therefore, not modelled. The identified potentials at NUTS 0 to NUTS 3 are available as an open dataset in order to make the data usable for others. In addition, the theoretical potentials are presented in a web application (https://datalab.dbfz.de/resdb/maps?lang=en, last access: 20 October 2023) to facilitate the scientific data communication.

Brosowski et al. (2016) described the difficulty in comparing different studies on biomass estimation due to missing standards and biomass categorisation. The study describes the methodology and scheme to define and categorise 77 by-products, residues and wastes in detail. The definition of by-products, residues and wastes is also applied in this study. The theoretical biomass potential is, according to (Thrän and Pfeiffer (2015) and Brosowski et al. (2020), considered as the maximum available biomass quantity under optimal management conditions. Since primary data on residues and by-products are often not available, a common method is the usage of so-called residue-to-product ratios (RPRs) (Brosowski et al., 2020; Scarlat et al., 2019; Weiser et al., 2014). In this approach, known production volumes are multiplied by an RPR factor, yielding an estimated quantity of a specific residue. Although the method is widely used, the applied factors are not standardized and can differ significantly. Hence, depending on the authors' or experts' choice, different RPRs are applied, leading to a wide range in biomass potential estimation over the different studies (Hamelin et al., 2019). Other factors increasing the uncertainty are the conversion of the theoretical biomass potential from fresh matter into other units such as dry matter (DM) or petajoule (PJ), because conversion factors such as water content or heat value also differ in the literature. The same is true for technical and other potentials. Depending on which restriction factors are included and where the thresholds are, the setting of the resulting potential can vary significantly (Brosowski et al., 2016). The focus of this study is primarily on model validation and extension in time, and, hence, no attention is paid to the comparison of the calculated biomass quantities, the conversion into other units and potentials or the further use of the data.

## 2 Materials and methods

Figure 1 gives an overview of in- and output data of the developed datasets. The first two rows show the input data sources used, their spatial scope and how the data have been combined to regionalise the biomass potentials. The last two rows show the resulting spatial output datasets. Where feasible, the time series were built for the period between 2010 and 2020, and the areas of the NUTS 2016 entities applied. However, due to alterations in the NUTS regions during the presented period, some data points from 2016 and earlier were recalibrated to ensure a uniform geographic representation across the entire time series. The approaches for the theoretical potential calculation have been adopted for the different residues as follows.

### 2.1 Biogenic municipal waste

#### 2.1.1 Theoretical potential

Regionalised biogenic waste data from households for each administrative unit were calculated by multiplying national-specific waste generation values per capita by the population value of the respective administrative area, using an approach similar to that employed in other studies such as Bellot et al. (2021) and Hamelin et al. (2019). In contrast to those two studies, however, a bio-waste allocation model which provides interpolated time series data over an 11-year period (2010–2020) while considering changes in the NUTS entities areas is proposed in this study.

Specifically, the model is built on data derived from the statistical office of the European Union (Eurostat). In detail, the sheets "env_wasgen" (eurostat, 2023c) containing and "demo_r_pjanaggr3" (eurostat, 2023d) were utilised. The sheet "env_wasgen" contains statistics for all EU-27 countries at NUTS 0 level for the European waste classification categories "*W091-Animal and mixed food waste*" and "*W092-Vegetal waste*" from different Nomenclature of Economic Activities (NACE) activities. In this study, however, we solely focus on the biogenic waste generated from private households. The W091 and W092 data generated from private households are summed to calculate the total amounts of generated biogenic waste. The Eurostat sheet "demo_r_pjanaggr3" provides yearly population data for all

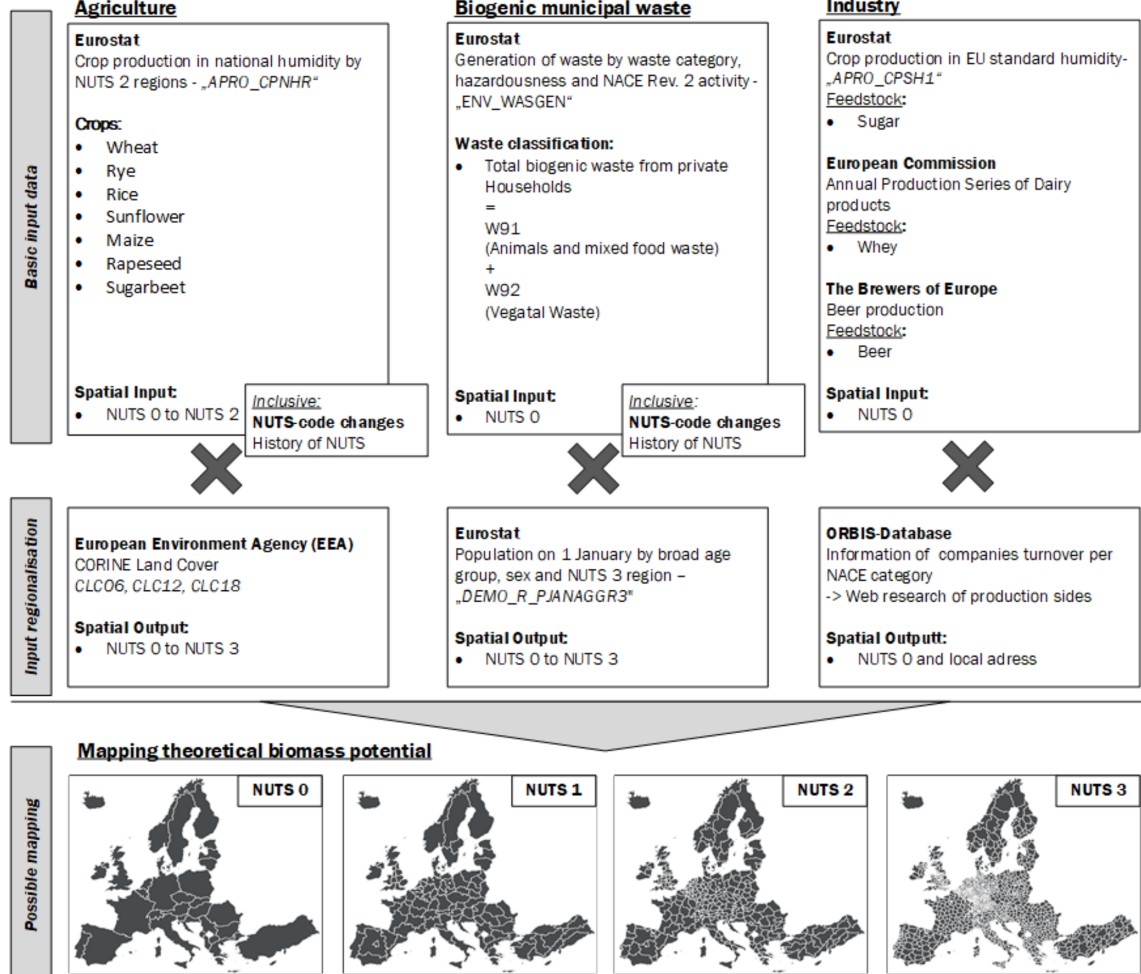

**Figure 1.** Overview of input and output data of the study 2.1 Biogenic municipal waste.

EU-27 countries on all four NUTS levels and was also utilised to create the model.

Before the calculations were performed, several data gaps in the primary data tables were addressed using interpola-
5 tion techniques. For example, European waste data are solely published every other year. The resulting gaps are closed by linear interpolation of two neighbouring data entries. Broader gaps of several years also occurred but were not interpolated in order to avoid extensive data manipulation. Missing values
10 in local population data (such as those at NUTS 3 level) are filled by calculating the differences between the incomplete population data and the total population values of the respective higher NUTS level (e.g., NUTS 2 level), and distribut-ing the derived differences proportionally to the missing data
15 fields.

The application of these interpolation techniques effec-tively addresses a significant number of data entry gaps, re-sulting in improved temporal availability of data on biogenic waste amounts from separate biogenic waste collection for
20 the years 2010–2020 at all NUTS levels.

To examine regional differences, the Member States were grouped into four geographical regions: Eastern Europe (Bulgaria, Czech Republic, Hungary, Poland, Romania and Slovakia), Northern Europe (Denmark, Estonia, Finland, Iceland, Lithuania, Latvia and Sweden), Southern Europe 25 (Croatia, Cyprus, Greece, Italy, Spain, Malta, Portugal and Slovenia) and Western Europe (Austria, Belgium, France, Germany, Ireland, Luxembourg and the Netherlands).

### 2.1.2 Data validation

To assess the accuracy of the predicted waste amounts, the 30 modelled data are validated against regional waste statistics. For this purpose, waste statistics from nine EU-27 Mem-ber States at varying regional resolutions were gathered. An overview of the compiled validation data is shown in Table 1.

The coefficient of determination ($R^2$) was used as a mea- 35 sure of goodness for the fit of the model, which was de-termined for the combined dataset on three regional levels (NUTS 1 to NUTS 3), as well as for individual countries.

**Table 1.** Overview of regional statistics on the quantities of biogenic waste generation from private households used to validate the model. In parenthesis: number of compiled data points of individual countries.

| Country | NUTS 1 | NUTS 2 | NUTS 3 |
|---|---|---|---|
| Austria | Yes (3) | Yes (9) | No |
| Germany | Yes (16) | Yes (36) | Yes (389) |
| Ireland | No | Yes (3) | Yes (8) |
| Italy | Yes (5) | Yes (21) | Yes (106) |
| The Netherlands | Yes (4) | Yes (12) | No |
| Poland | Yes (6) | Yes (16) | No |
| Portugal | Yes (2) | Yes (7) | Yes (18) |
| Slovakia | No | Yes (4) | Yes (7) |
| Spain | Yes (7) | Yes (14) | No |

In this country-wise analysis, countries with fewer than four data points were excluded to ensure that the $R^2$ could be calculated accurately.

## 2.2 Agricultural by-products

### 2.2.1 Theoretical potential

Seven agricultural by-products from feedstocks (maize, rapeseed, rice, rye, sugar beet, sunflower and wheat) listed in Eurostat database are included in the calculation of the theoretical biomass potential for the time series. The production data of the relevant feedstock is extracted from the Eurostat data table "APRO_CPNHR (eurostat, 2023b) on NUTS 0, NUTS 1 and NUTS 2 level. However, continuous data series of NUTS 1 and 2 regions were not always available in Eurostat due to spatial changes, missing data or non-adjusted data from Member States. Therefore, gaps are calculated with an approximation method using data from higher level or older spatial areas to regionalise the theoretical biomass potential (TP) and close data gaps on a spatial level. The agricultural area defined by CORINE Land Cover (European Environment Agency, 2019a, b, c) was used and combined with the production rate. Therefore, the share of the CORINE agricultural area ($A$) for NUTS level of interest ($i$) according to the year ($t$) and feedstock ($n$) is set in relation to the area of the next higher available NUTS level ($i - j$) CE2 and is multiplied with the production amount ($P$) and the RPR of the higher NUTS level (1). TS1

$$\text{TP}_{i,n,t} = \frac{A_{i,n,t}}{A_{i-1,n,t}} \times P_{i-1,n,t} \times \text{RPR}_n \quad \exists (i, n, t) \in (I, N, T).$$
$$(1)$$

Similarly to the approach of Bellot et al. (2021), the production volume is multiplied by the country-specific RPR of Scarlat et al. (2019) to calculate the theoretical biomass potential. For sugar beet leaves, the RPR from the German Federal Ministry of Food and Agriculture (Bundesministerium

für Ernährung und Landwirtschaft 2017) was used and applied to all countries. To enhance the dataset to a comprehensible time series from 2010–2020, CORINE Land Cover products are always connected to Eurostat data according to their reference timeline. In addition to arable land, Bellot et al. (2021) also consider heterogeneous agricultural land for downscaling to NUTS 3 level. We, on the other hand, only include the CORINE Land Cover classes 211 (non-irrigated cropland) and 213 (rice fields) in order to be more restrictive with regard to the possible NUTS-specific cultivated area.

### 2.2.2 Data validation

The modelled NUTS 3 production data are validated against the German national statistics, which provide data on harvested area (Regionalstatistik, 2022b) and regional specific yield values (Regionalstatistik, 2022a) for the year 2016 on NUTS 3 level. For each NUTS 3 region, the estimated production from CORINE and Eurostat was compared to the German production multiplied by the harvested area inside the NUTS 3 region and the corresponding yield. The resulting $R^2$ describes the accuracy of the model results. In addition, the calculated standard deviation (sd) gives an overview of the variation of the modelling.

## 2.3 Industrial residues

### 2.3.1 Theoretical potential

Data availability and sources are, if they exist, highly diverse in this category. Hence, no automated "fit for all" approach can be applied here. Input data for sugar production is retrieved from the Eurostat data sheet APRO_CPSH1 (eurostat, 2023a) for the full time series. Beer production is published by the European association of brewers (The Brewers of Europe, 2020) for 2012–2018. Following the RPR approach, production volume data for sugar and beer processing are multiplied with the specific conversion factor for the different residues. However, a range of conversion factors can be found in the literature. Therefore, only factors with documented measurements and plausible values have been considered; they are shown in Table 2. The range resulting from different RPRs for the same residue is reflected in the calculated biomass potentials as minimum and maximum values. Differently to agricultural-by-products the conversion factors vary depending on the technology and processes involved rather than on the geographical specifics. Since there was no further information available on these factors this aspect was neglected and the calculated average of minimum and maximum applied to all entities and points in time. Data on whey are used directly from Eurostat (European Commission, 2021, p. 66 TS2) and are available for 2010–2020.

All data are only available in NUTS 0. To achieve a regionalisation of the biomass potential, production sites are mapped using open data from industry associations, company websites and the Orbis company register database (Bu-

reau van Dijk, 2022). However, data on production volumes or the amount of residues are rarely shared by the companies and are, therefore, difficult to obtain. Therefore, the biomass potential cannot be regionalised by volume, but preference regions can still be visualised using the number of production sites per NUTS entity. Across the EU Member States 74 production sites of sugar were found and mapped. Dairies and breweries are slightly more difficult to map due to the high number of existing production sites. Therefore, in a first step the Orbis database was used to filter out the 50 companies with the highest turnover rate in Europe. Orbis only provides the location of the headquarters. Hence, the production sites of the 50 identified companies are searched utilising data from company websites and associations. Unlike sugar and beer factories, locations of dairy production sites are rarely disclosed by companies and production site numbers are much higher, resulting in no regionalisation of whey in this study.

### 2.3.2   Data validation

Due to the missing link between production sites and volumes no regionalisation could be carried out. Therefore, a validation is not needed either since there are no modelled data.

## 3   Results

### 3.1   Biogenic municipal waste

#### 3.1.1   Theoretical potential

Data on biogenic waste generation from private households, as reported by Eurostat, show a positive trend in total amount generated in the decade under study (2010–2020). The EU-27 Member States generated a total of 37.2 million tons of biogenic waste in 2020, a 68 % increase from 2010. Highly populated countries such as Germany, France, Italy, the Netherlands and Poland generated the highest amounts.

Western European countries collect the largest amounts of biogenic municipal waste due to high collection rates and population size. This group has seen ongoing growth in separate waste collection, with France and Austria reporting a more than doubled quantity of biogenic waste per inhabitant and year in the decade under study. With 128 kg per inhabitant and year, Austria reported the second-highest figure among EU-27 Member States. The rising population numbers in this group also contribute to the observed increase. Interestingly, Luxembourg reported only a fraction of its biogenic waste collection in 2020 (∼ 4400 t) compared to 2010 (∼ 67 300 t).

Collection figures have increased significantly in both Northern and Southern Europe, and Eastern Europe experienced a particularly pronounced increase, raising from 0.46 million tons in 2010 to 3.03 million tons in 2020, representing a 564 % increase. This is mainly due to the substantial growth in collection volumes in Poland, the most populous country of the Eastern European EU-27 Member States, which recorded the highest increase in the total amount collected, increasing its value by a factor of 20 in the decade under study.

Although Eastern Europe has seen positive trends in biogenic waste collection volumes, they still show the lowest figure in average quantity of biogenic waste collected per person and year. However, there is significant room for improvement in Southern Europe, too. The average amount generated by private households collected per person in 2020 was 34.5 kg in Eastern Europe and 38.8 kg in Southern Europe, while Northern Europe and Western Europe (excluding Luxembourg) collected on average 52.4 and 104.5 kg of biowaste per person and year, respectively.

#### 3.1.2   Data validation

At the NUTS 1 level, the $R^2$ for the combined dataset is 0.91 (average$_{\text{valid-data}}$ = 657 106 t; sd$_{\text{residuals}}$ = 204 032 t), indicating a strong fit between the model's predictions and the statistical data. For individual countries, however, a scattering of $R^2$ values can be observed. The $R^2$ values range from 0.64 (Italy) to 0.93 (Germany), with an average of 0.78. This suggests that the model is capable of accurately predicting waste production for larger regions.

At the NUTS 2 level, the $R^2$ for the combined dataset is 0.82 (average$_{\text{valid-data}}$ = 167 512 t; sd$_{\text{residuals}}$ = 78 135 t). For individual countries, the $R^2$ values range from 0.42 (Spain) to 0.80 (Italy), with an average of 0.63, which also indicates a good accuracy of waste production estimates for medium-sized regions.

At the NUTS 3 level, the $R^2$ for the combined dataset is 0.77 (average$_{\text{valid-data}}$ = 32 092 t; sd$_{\text{residuals}}$ = 17 470 t). Here, the $R^2$ values vary considerably from 0.02 (Slovakia) to 0.93 (Germany); the average is 0.62. It can be noticed that the accuracy decreases with increasing spatial resolution, while the range of the determined $R^2$ values becomes larger.

### 3.2   Agricultural by-products

#### 3.2.1   Theoretical potential

By integrating the NUTS code changes between 2010 and 2020, a complete time series of biomass potentials could be presented, as shown in Fig. 4. The highest theoretical biomass potential of 337 million tons in Europe (2014) resulted from the high production volume of sugar beet combined with high wheat production. Over time, the potential varied between 276 million (2010) and 337 million tons (2014). The variation is mainly driven by the available potential of wheat straw and sugar beet leaves in particular years, as these are the largest contributors to the total potential of agricultural residues. The maximum range of 61 million tons of theoretical biomass potentials occurred within 4

**Table 2.** RPR for industrial residues.

| Residue | RPR min | RPR max | Source | Ø RPR min used | ØRPR max used |
|---|---|---|---|---|---|
| Spent grains | 0.2 | 0.23 | Gupta et al. (2010) | 0.2 | 0.2 |
| Spent grains | 0.2 | 0.2 | Mussatto et al. (2006) | | |
| Spent yeast | 0.02 | 0.04 | The Brewers of Europe (2022) | | |
| Spent yeast | 0.02 | 0.04 | Avramia and Amariei (2021) | 0.085 | 0.11 |
| Spent yeast | 0.15 | 0.18 | Jaeger et al. (2020) | | |
| Molasses | 0.04 | 0.06 | MECAS (2016) | 0.04 | 0.06 |
| Beet pulp | 0.4 | 0.4 | Gaida (2013) | 0.45 | 0.45 |
| Beet pulp | 0.5 | 0.5 | Legrand (2015) | | |

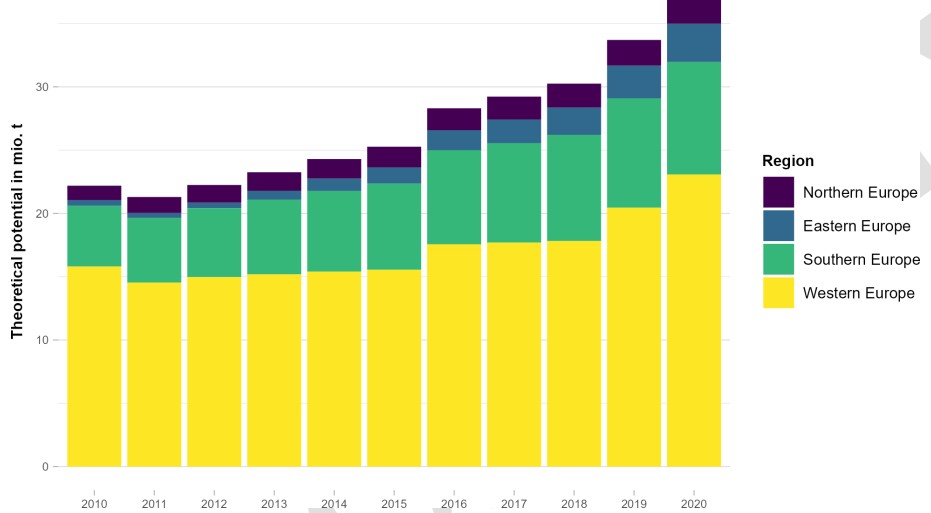

**Figure 2.** Theoretical potentials of biogenic waste from households in mio. t FM yr$^{-1}$ for the time series from 2010–2020. Sum of country values on NUTS 0 level, grouped by regional affiliation.

**Table 3.** Statistical evaluation of the data validation for the modelled data on bio-waste on NUTS 1, NUTS 2 and NUTS 3 levels.

| Country | NUTS 1 | NUTS 2 | NUTS 3 |
|---|---|---|---|
| Austria | 0.90* | 0.62 | – |
| Germany | 0.93 | 0.73 | 0.52 |
| Ireland | – | 0.98* | 0.96 |
| Italy | 0.64* | 0.80 | 0.82 |
| The Netherlands | 0.77* | 0.62 | – |
| Poland | 0.76 | 0.74 | – |
| Portugal | 1.00* | 0.63 | 0.77 |
| Slovakia | – | 0.45* | 0.02 |
| Spain | 0.68 | 0.42 | – |
| Combined | 0.91 | 0.82 | 0.77 |

Values marked with * are based on five or less data points.

years from 2010 to 2014. This shows the time dependence of biomass availability.

### 3.2.2 Data validation

Data validation was possible for four crops (wheat, rye, rapeseed and sugar beet) in Germany and was carried out by comparing the production amount from national statistics with our modelled production. As shown in the scatter plot of Fig. 5 and the corresponding $R^2$ values in Table 4, the agricultural crops wheat, rye and rapeseed had $R^2$ values between 0.61 and 0.95. These $R^2$ values indicate that the method of downscaling the production to regional levels using of CORINE Land Cover data was feasible. The corresponding (sd) in Table 3 shows that within certain NUTS regions the values between modelled and statistically reported production can differ significantly. The downscaling of sugar beet production by the CORINE Land Cover datasets leads to $R^2$ values of 0.30 (NUTS 2) and 0.20 (NUTS 3) and less valid results for the correlation between the modelled data

and the data from official German statistics. For sugar beet, the sd with respect to the average value is higher and shows that the local production value can deviate considerably.

### 3.3 Industrial residues

#### 3.3.1 Theoretical potential

The theoretical potentials of industrial residues in EU-27 are displayed in Fig. 6, which shows the theoretical potentials of the five industrial biomass residues. For datasets with a data range from minimum to maximum, only the minimum value is visualised. The bar plots clearly show the comparably high theoretical biomass potentials of whey and beet pulp. The amount of whey increased steadily from 37 to 44 mio. t FM yr$^{-1}$ over the 11-year time period with an exceptional high of 50 mio. t FM yr$^{-1}$ in 2019. The largest producer of whey in the EU is Germany with 14 mio. t FM yr$^{-1}$, followed by the Netherlands with 9 mio. t FM yr$^{-1}$, Poland with 7 mio. t FM yr$^{-1}$ and Italy and Ireland with 4 mio. t FM yr$^{-1}$. Nearly all countries show an increasing trend in whey production, especially Poland and the Netherlands. As explained, the mapping of hot-spot regions is not possible here.

The potentials of beet pulp show more variation from one year to another with a minimum of 43 mio. t FM yr$^{-1}$ in 2015 and a maximum of 60 mio. t FM yr$^{-1}$ in 2017. Eurostat data reveal that the three main sugar producers in Europe are France and Germany, with nearly 30 % production share, and Poland with more than 10 %. With the reform of the Common Agricultural Policy in 2013, the existing quota system on sugar was eliminated in 2017. The data show that this did not lead to a significant long-term change in biomass potential of residues from sugar production. Moreover, the production area decreased in Germany and Poland already in 2015 and 2016. In 2017, the area increased again in Germany to the former level and in Poland to even slightly more than before. A significant increase in residues from sugar production cannot be detected after 2017. Reasons for this might be a stronger influence of changing weather conditions combined with production area and the capacity of existing sugar factories. Along with beet pulp also molasses (minimum) varies. The numbers are less significant because the amount of residue accrued during production is much less. Overlaying the NUTS 0 beet pulp potentials with the number of production sites underlines the high sugar industry residue potentials of Germany, France and Poland from the Eurostat data. Comparing the map of potentials with amounts of factories per NUTS 0, 1, 2 and 3 shows the surplus of the method (Fig. 7). For countries with only one factory, like Sweden or Finland, a direct potential amount can be derived from the data. In other countries, such as France, there is only one sugar production region, indicating the location of biomass potentials.

Compared to the other three industrial residues, spent grains and spent yeast biomass potentials are rather low but very stable over the shorter time period of 2012–2018. The potentials of spent grains (minimum) ranges between 6700 and 7100 t FM yr$^{-1}$ and for spent yeast (minimum) between 2800 and 3000 t FM yr$^{-1}$. According to the The Brewers of Europe (2020) Germany is the biggest producer of beer with nearly 30 % of EU production. This is followed by Poland, Spain and the Netherlands with about 10 % each. Mapping the 50 most important companies reveals a high concentration of production sites in Germany and the Netherlands, although they are widely dispersed. Spain, Poland, Denmark, Ireland and Sweden show regional hot-spot areas. The production in these countries is smaller, but the concentration of residues may be higher. A count of breweries shows that the highest concentration of breweries in NUTS entities are in Arr. Halle-Vilvoorde with five followed by Bas-Rhin, France and Munich, Germany, with three factories each.

### 3.4 Cross-sectoral analysis

#### 3.4.1 Theoretical potential

The estimated biomass potentials of all three categories show dynamics over the time period analysed. Looking at biogenic municipal waste, the total biomass potential grew from its lowest amount of 21 mio. t FM in 2011 to 36 mio. t FM in 2020 with big steps in the past 4 years. This is a difference of 71 % in only 9 years. Agricultural by-products show more variation of theoretical biomass potential between each year. Having the highest year in 2014 with 337 mio. t FM and the lowest 2010 with 276 mio. t FM, means a decrease of 18 %. The year 2014 was followed by 2015 and 2016 with a theoretical biomass potential of only 300 mio. t FM and 298 mio t FM, which is a decline of nearly 40 mio. t FM. More surprising are the dynamics in the industrial residues. The biomass potential decreased in 2014 to 2015 by 12 % in 1 year and increased in 2016 to 2017 by 17 % in the other year. Figure 8 shows the overall estimated biomass potential for all 13 wastes, by-products and residues per square kilometre, which gives another picture of the biomass density in one country. The three maps are chosen as 2010 being the first year of the time series and also the lowest year for biomass availability with 384 mio. t FM. In contrast, 2017 is the year with the highest available biomass amount with 469 mio. t FM, and 2020 is included as the last year of the time series with 427 mio. t FM.

With increasing data volume and continuous tempo-spatial expansion, databases need to be based on automatisation tools, such as in this study. Calculations were performed, using Eurostat data, in R statistical programming language and through direct data acquisition from Eurostat's API, allowing for automatic updates and ensuring the reproducibility of the results. However, automating the process can be disrupted by changes in the primary data, e.g. revisions in regional NUTS

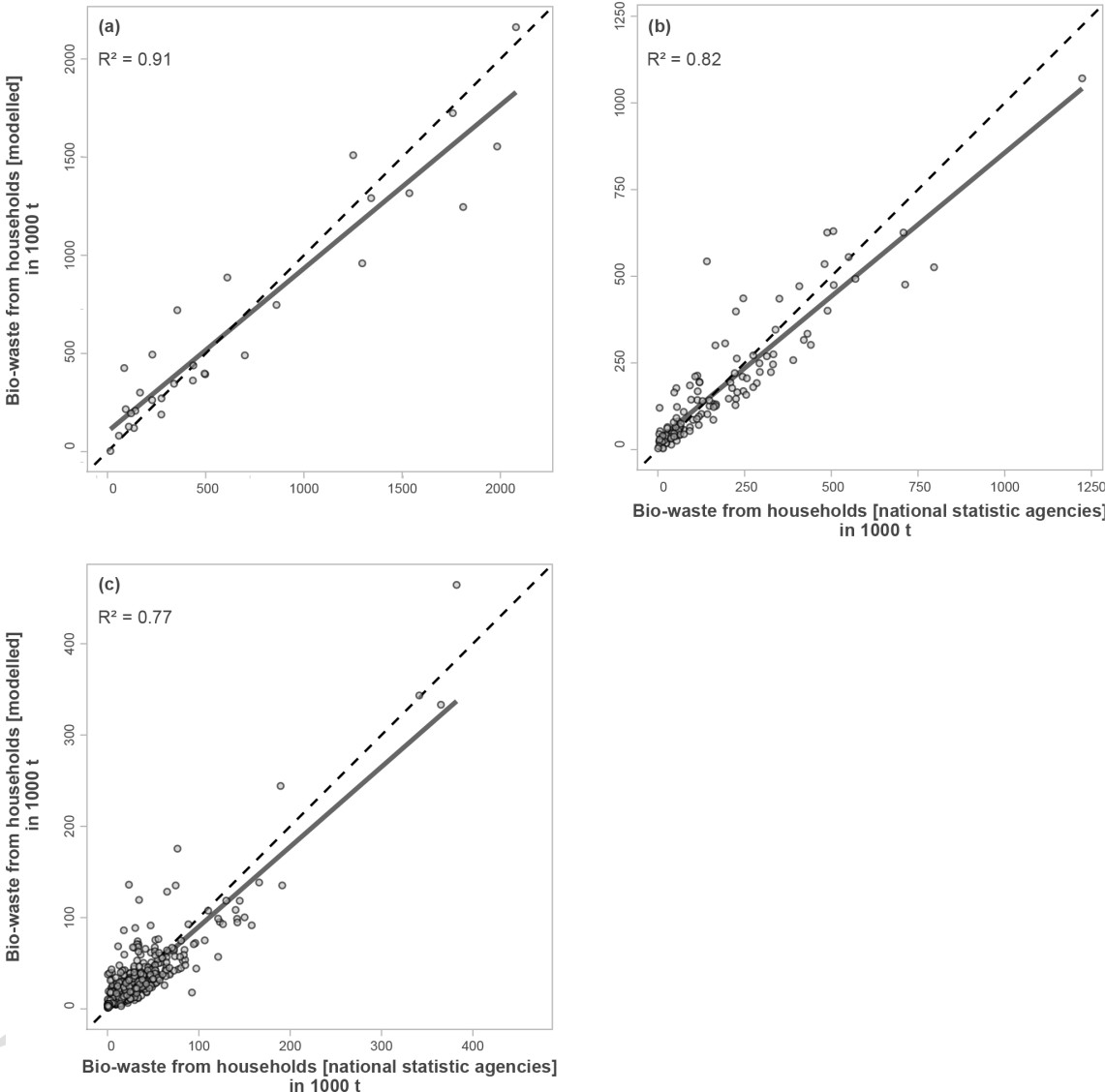

**Figure 3.** Correlation of modelled biogenic waste data and statistical data from statistical agencies in nine European countries at three different NUTS levels. **(a)** NUTS 1 level, **(b)** NUTS 2 level and **(c)** NUTS 3 level. Solid line: linear regression line; dashed line: 1 : 1 relation.

**Table 4.** Statistical evaluation of data validation for the agricultural sector at NUTS 2 and NUTS 3 levels in Germany.

|  | NUTS 2 | | | NUTS 3 | | |
|---|---|---|---|---|---|---|
|  | $R^2$ | $sd_{residuals}$ [t] | $average_{valid\_data}$ [t] | $R^2$ | $sd_{residuals}$ [t] | $average_{valid\_data}$ [t] |
| Wheat | 0.90 | 191 049 | 650 698 | 0.71 | 46 560 | 78 176 |
| Rye | 0.95 | 36 572 | 97 001 | 0.61 | 15 037 | 17 483 |
| Rapeseed | 0.87 | 61 108 | 128 290 | 0.80 | 9709 | 17 748 |
| Sugar beet | 0.30 | 647 959 | 78 150 | 0.20 | 140 784 | 140 011 |

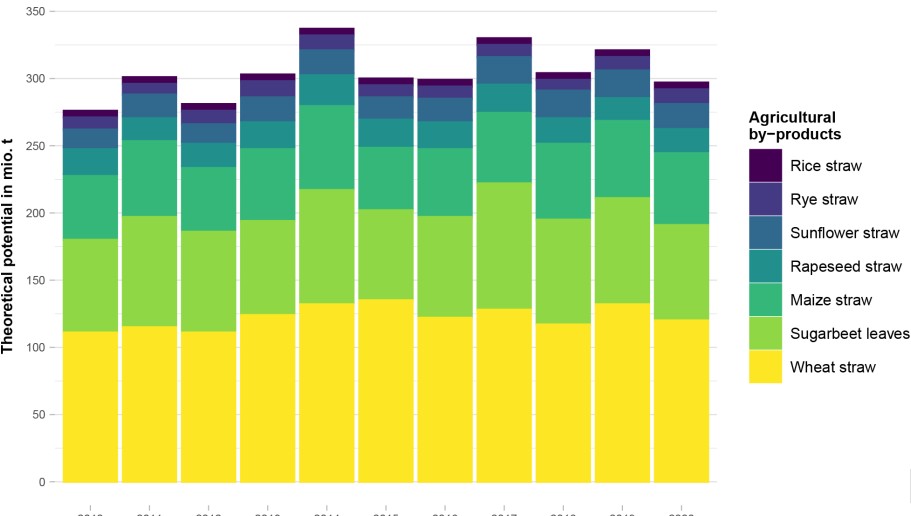

**Figure 4.** Theoretical potentials of agricultural by-products in mio. t FM yr$^{-1}$ for the time series from 2010–2020. Sum of all EU-27 countries on NUTS 0 level, grouped by biomass type.

definitions or the overall structure of the table provided by Eurostat, which must, therefore, be regularly monitored and possibly debugged. The open data results can be used instantly to calculate other products or visualise high potential biomass areas and time series as demonstrated in the DBFZ webapp. All results on NUTS 0 to NUTS 3 from this study are visualised here with further analytics and an open API.

## 4   Discussion

For biogenic municipal waste, the regionalisation approach based on national average waste generation rates and population data resulted in estimates of the amount of separately collected biogenic waste from private households for the NUTS regions of the EU. In contrast to comparable studies, such as Bellot et al. (2021) and Hamelin et al. (2019), a validation of the regionalised data was performed to assess the accuracy of the modelled estimates. For this purpose, regional statistical data from several EU Member States were compiled. When correlating the modelled and the statistical data, $R^2$ values ranging between 0.77 on NUTS 3 level and 0.91 on NUTS 1 level were found. This suggests an overall high accuracy and reliable estimation of the generated amounts of biogenic waste at all three levels of regionalisation. However, weak or no correlations between the modelled and statistical data were observed for certain countries, e.g. Spain and Slovakia. This may imply that significant deviations exist between national averages and actual values of individual regions in these countries, leading to higher errors in the estimation. This is particularly challenging for countries with a low number of statistical entities, such as Slovakia, which consists of only seven NUTS 3 regions. In such cases, marked deviations occurred in the validation of the data, which could reflect differences in population behaviour or

policy implementation status across regions within a country. Additionally, differences in the enforcement and implementation of waste management policies and regulations may also contribute to the observed deviations. Therefore, results on NUTS 3 level must be taken with caution, especially in countries where nation-wide implementation of separate collection systems is of immature status and hence prone to heterogeneity across regions.

The estimation of agricultural by-products was validated by comparing modelled results of crop production with regional statistical data from Germany on NUTS 2 and NUTS 3. The validation demonstrated overall good accuracy in the estimated straw residues declining with higher regionalisation. With an $R^2$ of 0.80 for rapeseed and 0.71 for wheat straw, the most common crop residues in Europe, still show very good results on NUTS 3 with the regionalisation method used. Nevertheless, the standard variation reveals that regionally high deviation can occur. The accuracy of modelled sugar beet leaves shows low correlation on NUTS 2 and NUTS 3. One explanation for this is that sugar beet requires specific soil conditions, which are not available everywhere. Therefore, the area under sugar beet is limited to selected areas within a higher NUTS level, so that an area-wide distribution of sugar beet using the CORINE Land Cover class 211 (non-irrigated arable land) makes the downscaling to a lower NUTS level less accurate. This shows that the regionalisation method based on the share of agricultural area is not applicable for smaller and very regional crops. It is also important to note that external validation is subject to the uncertainties inherent in the official statistics used. These uncertainties may arise, for example, if the quantities reported are only approximate estimates from local or sub-national authorities. This also accounts for the external validation of the modelled es-

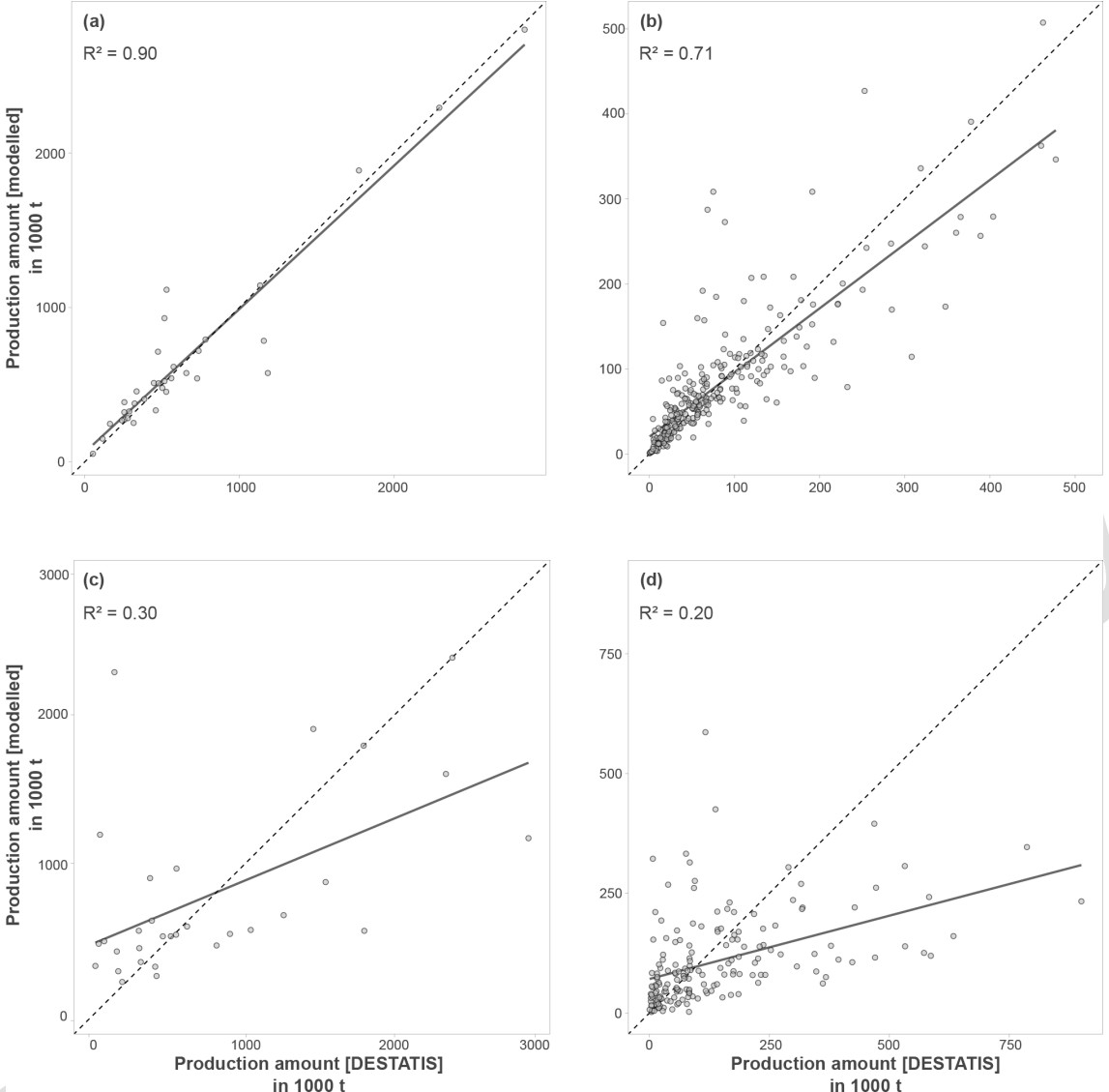

**Figure 5.** Correlation of modelled agricultural production amounts with official German statistics (Regionalstatistik, 2022a, b). **(a)** Wheat for NUTS 2 level, **(b)** wheat for NUTS 3 level, **(c)** sugar beet for NUTS 2 level and **(d)** sugar beet for NUTS 3 level. Solid line: linear regression line; dashed line: 1 : 1 relation.

timates of agricultural by-products, as well as the estimates of biogenic household wastes.

For industrial residues, only a partial regionalisation of biomass potentials was feasible due to mostly confidential production data (Patricio et al., 2020; Song et al., 2017). The gap of input data for biomass estimation is also reflected by the low number of studies in this field. Some studies use socio-economic indicators, waste statistics and company registers to estimate the general amount of biogenic residues (Patricio et al., 2020; Caldeira et al., 2019), but on an individual residue and wider scale, studies are very limited. Moreover, company registers only provide data at headquarters level, which makes it difficult to break down the data to the actual production sites and assess their geographical distribu-

tion. Therefore, the spatial approximation approach used was merely a first step to regionalise industrial biomass potential, but further research and open data are needed to combine it with other proxy data in order to estimate the spatial availability of individual biomass potentials. Despite these limitations, the visualisation illustrates that regions with higher biomass potentials derived from industrial activities can be identified in most cases.

With regard to the total biomass availability of the time period studied, it was shown that trends depend on the geographical location. For example, the biomass potential increased in Europe by 86 mio. t FM between 2010 and 2017. Looking at the map its noticeable that this effect is driven by middle and eastern EU countries. The controlling effects

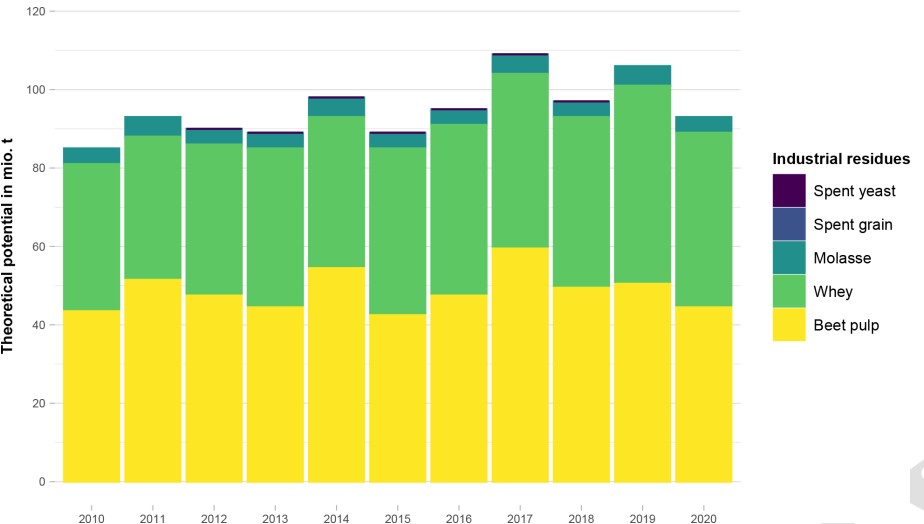

**Figure 6.** Theoretical potentials of industrial residues in EU-27 in 2010–2020. Sum of all EU-27 countries on NUTS 0 level, grouped by residue type.

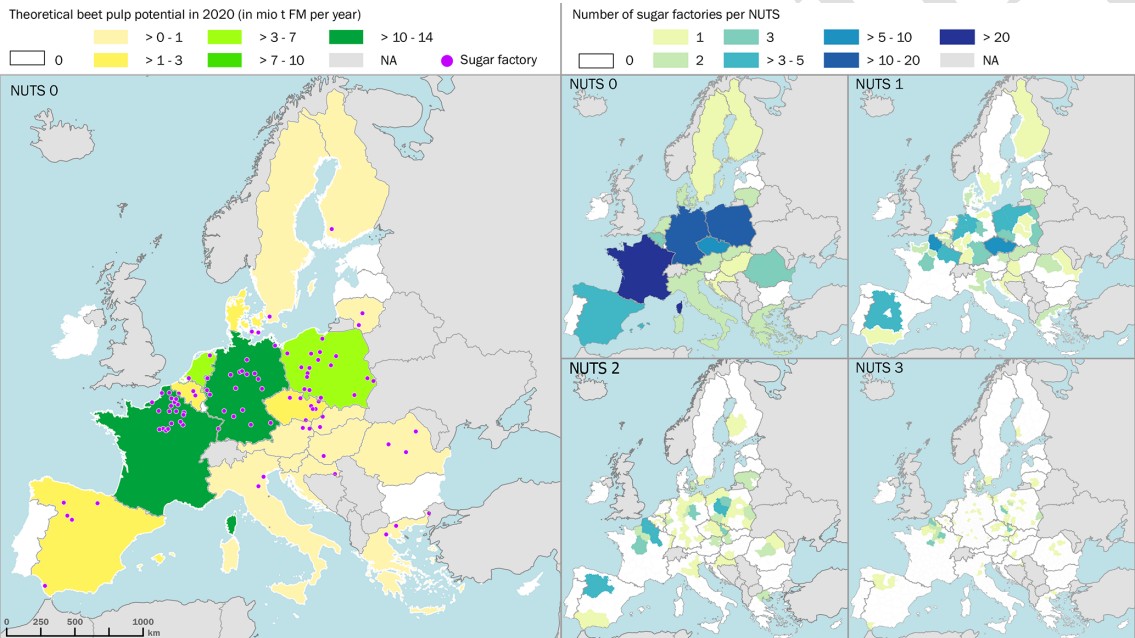

**Figure 7.** Theoretical biomass potentials of beet pulp and location of sugar factories.

of these curves are different in each sector. Some effects are long term and slower than the change in the WFD or the elimination of sugar quotas, while others, such as weather conditions, have an immediate effect every year and strongly vary over the NUTS 3 entities compared to the average NUTS 0, which was also shown previously by Brosowski et al. (2020). With double-digit changes in biomass potential from one year to the other, it can be concluded that an estimation of available theoretical biomass potentials based on a single reference year is not sufficient to understand the national and re-

gional biomass or estimate future availability. Additionally, in this study, the area changes in NUTS 2 and 3 over time were considered to ensure comparability between the years analysed and the amounts estimated.

As discussed, different methods, input data and reference years, and also the definition of theoretical biomass potentials, make the comparison between studies rather difficult. Especially at regional level, the estimation error can be high. This study showed for the biomasses analysed that validation on the regionalised data and long-term data monitoring

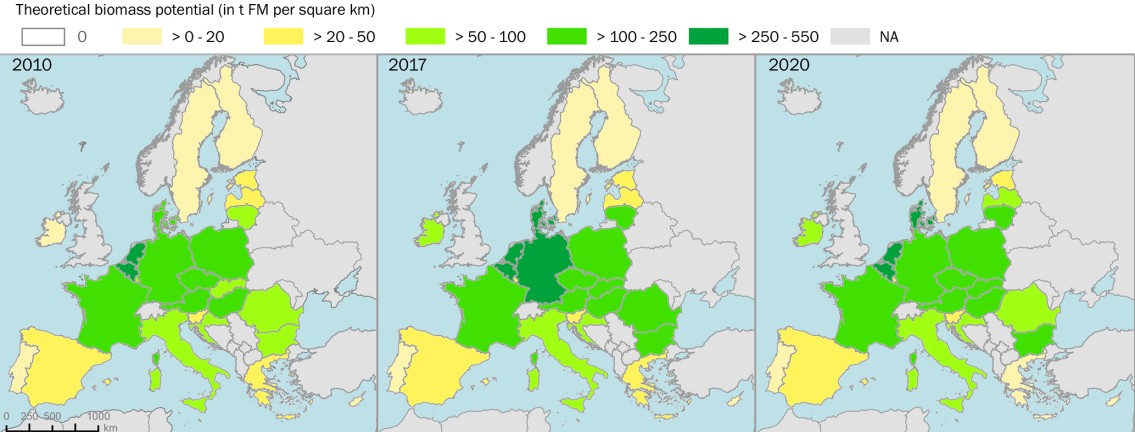

**Figure 8.** Theoretical biomass potentials from the presented biomasses in 2010, 2017 and 2020. Cross-sectoral sum in relation to country specific area.

is necessary to understand biomass availability. To achieve this, the data situation must be improved on the one hand and the survey methods on the other. This study was limited to 13 wastes, by-products and residues and should be extended to other biomasses such as animal by-products and residues from the forestry and timber industry in a next step, as these are the most relevant biomasses in Europe, as shown by Karras et al. (2022). Besides the extension to further theoretical biomass potentials, the inclusion of additional countries, years and units would also increase the potential use of the dataset. This is especially true for the conversion to PJ or DM as standard units for comparing the bioenergy and material use potentials, respectively, of the individual biomasses. However, as discussed at the beginning, this step would increase the model uncertainty and bias. Additionally, with recent remote sensing novelties, high-resolution cropland area products, such as from d'Andrimont et al. (2022) or Blickensdörfer et al. (2022), have now become available and need to be considered for regionalisation of agricultural by-products in Europe. The biggest challenge lies in the data collection of industrial residues, and this must be improved through regulatory requirements. With other improvements in current satellite imagery and classification algorithms combined with datasets such as FAOSTAT, advanced regional biomass estimates may be achieved in the future for agricultural by-products and forestry residues on a global scale. However, when using input data from administrative units, spatial changes over time must be taken into account when constructing time series, as well as locally specific RPRs. Due to the already discussed limitations of input data for Europe, regionalisation of other biogenic residues, e.g. from livestock, biogenic municipal waste or industry, will remain a challenge for a continuous global monitoring.

## 5 Data availability

The dataset that supports the findings of this study is openly available in OpenAgrar at https://doi.org/10.48480/g53t-ks72 (Günther et al., 2023). Data visualisation of the above dataset can be found at https://datalab.dbfz.de/resdb/maps?lang=en (last access 20 October 2023). TS3

## 6 Conclusion

This study provides valuable insights into the regional distribution and temporal trends of theoretical biomass potentials of 13 different biogenic wastes, by-products and residues in the EU-27 between 2010–2020. The study revealed the strengths and weaknesses of currently available primary data and biomass estimations. It was shown that data regionalisation works well in general. For biogenic household waste, the combined accuracy reached $R^2$ values of 0.91 (NUTS 1), 0.82 (NUTS 2) and 0.77 (NUTS 3). For agricultural by-products, average $R^2$ values of 0.76 (NUTS 2) and 0.58 (NUTS 3) were reached. However, it was also shown that the accuracy of the data can vary highly in NUTS 3 entities or, in general, with crops with a small overall cultivation area. Data of industrial residues lack availability and quality, which have made regionalisation difficult on a European scale. The approximation approach using production sites is a first step but is only possible if the industry products are directly correlated to the residues. With more complex industries such as dairy production, this approach is not feasible.

Consistent input data are important for building time series and hence trend analysis. With the correction of NUTS, area changes and gap filling of missing data, these time series are provided as an open source dataset (Günther et al., 2023). Variations in biomass availability are connected to the different NUTS levels and can be visualised. This helps us to understand regional and local trends as highlighted for

2017, which was the year with the highest amount of biomass over the studied time period, but biomass increases can be seen mainly in Central and Eastern Europe. The time series in NUTS 3 allows us to identify exceptionally low or high years, which also improves the forecasting of the theoretical biomass potential in the future. Providing an open access dataset and an online visualisation tool for temporal and spatial differentiation of the theoretical biomass potentials for the studied residues over several years is a step forward towards a reliable, continuous monitoring system. Using the dashboard provides immediate support for policymakers and investors. The dataset also supplies a valuable data product to other models, such as climate change mitigation, economic or energy models which also reduce the uncertainty of their part relying on long-term time series. The structure of the database supports not only the direct use of the data by following the findability, accessibility, interoperability, and reusability (FAIR) principles but also the inclusion of further biomasses. The need for such a data product in the growing EU bioeconomy sector is clearly given.

**Author contributions.** SG was responsible for conceptualisation, project administration and draft preparation of the paper. SG, TK and SS conducted the data curation, methodology, analysis, visualised data and wrote the final paper. TK and SS carried out the validation. FNdT and DT were responsible for the supervision and paper review.

**Competing interests.** The contact author has declared that none of the authors has any competing interests.

**Disclaimer.** Publisher's note: Copernicus Publications remains neutral with regard to jurisdictional claims made in the text, published maps, institutional affiliations, or any other geographical representation in this paper. While Copernicus Publications makes every effort to include appropriate place names, the final responsibility lies with the authors.

**Financial support.** This project has received funding from the Bio Based Joint Undertaking (JU) under grant agreement no. 887115. The JU receives support from the European Union's 2020 research and innovation programme and the Bio-Based Industries Consortium. TS4

**Review statement.** This paper was edited by Dalei Hao and reviewed by Matthew Langholtz and Xin Zhao.

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

## Remarks from the language copy-editor

## Remarks from the typesetter