# Peer review of "Temporal and spatial mapping of theoretical biomass potential across the European Union"

_Earth System Science Data, 2023_

## Referee Comment (RC2)

500

[referee-annotated manuscript omitted]

---

## Author Comment (AC1)

| Comments from referees (RC) #1 https://doi.org/10.5194/essd-2023-179-RC1 | Authors Response and changes in the manuscript |
|---|---|
| The study, "Temporal and spatial mapping of theoretical biomass potential across the European Union," extends the existing literature to construct a time-series dataset of potential biomass from major sources in the EU, and downscaling from country-level data to finer scales was made where possible. Overall, I (as an agricultural economist and integrated assessment modeler) find the data could be of important use. The paper is well-structured, but the writing and analysis can be significantly improved. In many places, additional details and clarification are needed. The contribution of the data/study, the usefulness of the data, and the impact of the analysis are not well-discussed. Authors may consider expanding the introduction and/or discussion sections to communicate why/how the data could be needed and used. Here are my detailed comments/questions. | Thank you very much for your detailed response and constructive suggestions to improve this study! Indeed, we agree with the general suggestions and have included them in the new manuscript. The Introduction, Analysis and Discussion have been expanded to address the comments and the general language has been clarified. However, in order to comply with the journal's scope on the description of the dataset and methods, we have kept the focus on this part with further details and clarifications. Nevertheless, additional examples of the need for this type of data and validation have also been included to provide examples for further use. |
| Based on the abstract, intro, or conclusion, I am not convinced that the data is important and useful. E.g., why expanding the biomass potential data temporally and improving the quality of the data important, besides understanding the trend and regional heterogeneity? | We have extended the introduction, discussion and conclusion sections to ensure that we cover this important point better.

We have added line 51ff. which additionally describes the missing monitoring tools at the different spatial scales for MSW. Lines 59ff. have also been added to extend the current coverage of available datasets at EU level, their spatial and temporal coverage and the gaps we see. For example, studies based on a single reference year that also provide future biomass potential can, as our analysis showed, vary significantly depending on which year the analysis is based on. In addition, most studies and projects are not peer reviewed. The need for this qualitative time series of available and recorded data is now more clearly demonstrated. The added summary from line 70 explains the potential use in the bioeconomy with a focus on the products developed in CAFIPLA. Additionally, in line 480ff we explain the usefulness for other products such as different models and that the continuous, open and machine-readable structure makes this dataset so useful. |
| In the recent literature of climate change mitigation scenarios (e.g., those projecting future bioenergy production using | We have included this valuable comment in line 480ff of the new manuscript. As explained, our studies and the journal's scope focus on the modelled data. |

| | |
|---|---|
| integrated assessment models such as GCAM, GLOBIOM, IMAGE, etc., see the recent IPCC WG3 report or the scenario database), almost all scenarios require large scale residual biomass to be used in near future. So those models indeed included methods of calculating the biomass potential (and more importantly, a supply curve) for all regions. I believe there is an important need in that literature to utilize your data. Some discussion could be useful, e.g., comparing approaches. | However, a comparison of different scenarios focusing on their data sets and assumptions could be an interesting study. |
| A broader literature review/discussion could be useful. | To improve our work in this regard, we added several literature sources with the works of, for example, Brosowski et al. (2016), Bell et al. (2018), Iglesias et al. (2020), Siegfried et al. (2023) and datasets from S2BIOM and ENPRESO. The discussion has been slightly expanded, with a focus on current gaps and future uses of the data at several points in the paper. |
| I understand the study tries to construct and connect the biomass data from three sources, Ag, MSW, & industry. However, what is the big picture? E.g., how important are the sources you didn't consider? E.g., how important are other sources like forestry/milling residues, other crops & industries. | The scope of CAFIPLA has been limited to the biomasses mentioned because of their technical and legal suitability for the technologies developed in the project. However, the types of biomass included in the study are determined by the context of the CAFIPLA project. Straw and municipal solid waste are two of the top 10 biogenic residues, by-products and wastes in Europe in terms of their technical potential, according to a literature review by Karras et al. (2022). Outside the context of the CAFIPLA project, it will be useful to extend our model to other biogenic residues, by-products or wastes, such as forest residues or manure, as these biomasses have great potential in Europe. For the units, we believe that the conversion increases the error, as factors, thresholds and restrictions also vary according to the authors' choices. A more detailed explanation is added in line 120ff. |
| You discussed the total potential in section 3.4, which I find useful. However, in aggregation, unit matters. Were you trying to add up biomass across sources with different water content? I suggest using dry matter tons and additional conversion to energy units, e.g., EJ/MJ could be extremely useful. | |
| Some clarifications on agricultural "by-product" and residues could be useful, and maybe also primary vs. secondary (industrial?). For example, it seems for maize, only strew in included, but not cob or other residues? | As mentioned above, in this first dataset we focus on the 12 residues required for the CAFIPLA project. The definitions of agricultural by-products, residues and wastes follow the classification of Brosowski et al. (2016). For clarification, a line has been added in the manuscript in 111ff. |
| Also, some discussion of the gap between potential biomass and the harvestable fraction could be useful. | Indeed, this discussion is very important for the amount of biomass available, biodiversity, soil health, ecosystem services and much more. However, we see that this discussion has been going on for decades without any final conclusions or agreements, nor any real legal restrictions. This paper focuses on the theoretical biomass potential as the "first" product of all other biomass potentials. The assessment of the amount of harvestable biomass is part of the technical |

| | and sustainable biomass potential. As explained, much depends on the author's choice of which constraints and thresholds are respected and thus would change the amount of biomass. To claim what is used and what is not is not the aim of this paper and it is left to the dataset user to make a responsible choice if other biomass potentials are needed. |
|---|---|
| Line 125, by model, do you refer to the downscaling model shown in Fig. 1? | We have revised the sentence "To assess the accuracy of the predicted waste amounts, the modelled data are validated against regional waste statistics. For this purpose, waste statistics from nine EU-27 Member States at varying regional resolutions could be gathered." To clarify. |
| Line 145, the equation is not very clear to me. Should it be RCR? | You are right, RPR is the more general term to express the proportion of residues from a main product. In the context of agricultural products, RCR is also used. According to our introduction, where we introduced RPR (line 86f.), we will continue the naming also for agricultural residues. |
| I think residue to crop ratio (RCR) should be discussed. Is it the same across places and years? | Thank you for that comment. We have revised the text and added more information to clarify.
Line 196ff of the new manuscript:
"Similar to the approach of Bellot et al. (2021), the production volume is multiplied by the country-specific RPR of Scarlat et al. (2019) to calculate the theoretical biomass potential. For sugar beet leaves, the RPR from the (Bundesministerium für Ernährung und Landwirtschaft (2017) was used and applied to all countries. " and line 223ff:
"Different than agricultural-by-products the conversion factors vary depending on technology and processes involved rather than on the geographical specifics. Since there was no further information available on these factors this aspect was neglected and the calculated average of minimum and maximum applied to all entities and points in time." |
| RPR was discussed, but Table 2 was not clear. Please make it independent | You are right, the header was not very clear. We have changed it to "ø RCR min used". |
| I agree the validation is useful. However, please note that it is only partial identification/validation. The external validity is not guaranteed. Source data quality is always the most important. | In fact, the validation data also contain errors and uncertainties. Unfortunately, this cannot be excluded. We pointed out the number of validation points and tried to find as many national and regional statistics as possible within the scope of the project. The paper therefore also discussed the reliability of the data used, with the conclusion that data collection and methods need to be greatly improved and standardised. In line 425f of the new manuscript a sentence addressing that issue has been added. |
| Similarly, I guess your approach may also be applied to downscaling the biomass | We believe that the data and methods can indeed be useful for further modelling. However, as we have |

| | |
|---|---|
| data produced by IAM in future periods, subject to some additional assumptions. | shown, the results at NUTS 3 level are highly dependent on the amount and characteristics of the individual biomass. |
| I guess it is also related to the use of the data. In many cases, NUTS1 or NUTS2 are enough, where you have better data. | Thank you for this comment. You are right that high level decisions, especially policy strategies including biomass use, are fine with NUTS 1 or 2. When it comes to concrete test cases, building new production sites and regional supply chains, NUTS 3 is very important. Transport costs become more important and for input biomasses, but also for products with higher water content, transport distance is an important factor. |
| More importantly, I think it would be extremely useful to discuss future work or the potential for the continuation of the work. For example, I wonder if the code was written in a flexible way for one to update the data, e.g., when new data (after 2020) become available. In my opinion, it is important to make the data/processing/update "live" since the data may become increasingly useful. | We also think a continuous database should be the standard to monitor biomass availability and provide a continuous data product for other application. To clarify your comment, we added some lines in the new manuscript (390ff): "Calculations, using Eurostat data, were performed in R statistical programming language and through direct data acquisition from Eurostat's API, allowing for automatic updates and ensuring the reproducibility of the results. However, automating the process can be disrupted by changes in the primary data, e.g. revisions in regional NUTS definitions or the overall structure of the table provided by Eurostat, which must therefore be regularly monitored and possibly debugged.".
 However, hindrance can be found in source data, e.g. changes in NUTS organisation or data structure. |
| FAOSTAT has some supply-utilization data that | Unfortunately, this sentence was not complete and we did not manage to imagine where it would lead. |
| In the final data you produced, did you use the data you produced, or you used the source data, where applicable, you used in the validation? | We considered publishing a combined dataset, but decided to provide modelled data only for methodological consistency. |
| Was agricultural production increasing over time? Why ag residues are not changing much over time. I would appreciate discussion of the potential drivers of the trend. | Thank you for your comment. As you can see in Figure 4, the potential varies over the years with + and - 10% around the average of 304 Mt[FM]/a.  There hasn't been a clear increase in the theoretical biomass potential over these eleven years. One reason for this may be that individual weather conditions, as a combination of temperature, sunshine and rainfall distributed throughout the year, affect the yield of the main product and therefore also the yield of agricultural by-products and residues. However, a detailed analysis of weather and yield conditions in the relevant years wasn't part of the scope of the study.
 In addition, we have added a line on this in 301ff. |
| Any thoughts on the residues of other crops/livestock products? | As mentioned above, some other residues, wastes and by-products are also of interest. With regard to the availability mentioned by Karras et al. (2022), manure and forest residues are of particular interest. |
| Line 404, what happened in 2017? | Figures 4 and 5 show good harvests of sugar beet and wheat. As a result, the theoretical biomass potential has |

| | increased. The reasons for this are beyond our scope and require deeper analysis, e.g. climate and political decisions. |
|---|---|
| | |
| **Comments from referees (RC) Matthew Langholtz** https://doi.org/10.5194/essd-2023-179-RC2 | **Authors Response and changes in the manuscript** |
| Valuable assessment of select waste and agricultural processing residues at various scales in the EU. Good job assessing scale-specific data uncertainty. Some questions and suggestions follow. Other copy-edit suggestions for your consideration tracked in the PDF. | Dear Mr. Langholtz, thank you very much for your review. Your comments and suggestions online and in the PDF were very much appreciated and helped to improve this study from the authors' point of view. We have revised the paper to be clearer and more specific. |
| Line 41: "Data on MSW streams indicate that landfill declined from over 60 % to 24 % over the last three decades." Percent of what? I think this is percent of MSW landfilled but this is not clear. | The sentence has been revised. See line 42 of the new manuscript: "Data on MSW streams indicate that landfilling has decreased from over 60% of MSW treatment to 24% over the last three decades.". |
| Line 43: "with each an increase of over 10 %" percent of what? | The sentence has been revised. See line 43ff of the new manuscript: "This is mainly achieved by increasing the rate of material recycling using composting and digestion of degradable wastes and incineration with an increase of each of the two waste treatment streams of ". |
| Line 174: "For almost none industrial food production sites individual production values can be found." Not clear, please rephrase. | The sentence has been revised. See line 231f of the new manuscript: "However, data on production volumes or the amount of residues are rarely shared by the companies and are therefore difficult to obtain". |
| Line 185: Some more explanation is suggested to help the reader understand why a validation is not needed. | The sentence has been revised. See line 244 of the new manuscript: "Therefore, a validation is also not needed since there are no modelled data.". |
| Line 210: Define mio. t [FM] at first instance in caption and first instance in text. Acknowledging the EU perspective, I don't think this is universally standard, and I don't recognize "[FM]". | The sentence has been added to clarify. See line 103 of the new manuscript: "The theoretical potential is expressed in units of specific mass and in terms of fresh matter [FM].". |
| Line 298: "The production site mapping of the 50 biggest companies show accordingly a high density in Germany and the Netherlands although very spread." Suggest clarify wording. | The sentence has been revised. See line 360f of the new manuscript: "Mapping the 50 most important companies reveals a high concentration of production sites in Germany and the Netherlands, although widely dispersed.". |
| Figure 7: change "Amount…" to "Number…" | The figure has been changed. In the delivered PDF it was already correct only in the manuscript it was not updated. |
| Line 313: "Agricultural by-products vary more in between each year." Clarify wording | The sentence has been revised. See line 377f of the new manuscript: "Agricultural by-products show more variation of theoretical biomass potential between each year.". |
| Line 316: "Decreasing in 2014 to 2015 by 12 % in one year and increasing in 2016 to | The sentence has been revised. See line 381 of the new manuscript: "The biomass potential is decreasing in |

| 2017 by 17 % in the other year." Incomplete sentence | 2014 to 2015 by 12 % in one year and increasing in 2016 to 2017 by 17 % in the other year.". |
|---|---|
| Line 320: "2017 is the year in contrast has with 469 mio. t [FM] the highest available biomass amount and2020 is included as the last year of the time series with 427 mio. t [FM]." Clarify wording. | The sentence has been revised. See line 385f of the new manuscript: "In contrast 2017 is the year with the highest available biomass amount with 469 mio. t [FM] and 2020 is included as the last year of the time series with 427 mio. t [FM].  ". |

**References**

Bell, J., Paula, L., Dodd, T., Németh, S., Nanou, C., Mega, V., and Campos, P.: EU ambition to build the world's leading bioeconomy-Uncertain times demand innovative and sustainable solutions, New biotechnology, 40, 25–30, https://doi.org/10.1016/j.nbt.2017.06.010, 2018.

Bellot, F.-F., Horschig, T., and Brosowski, A.: Quantification of European Biomass Potentials, 2021.

Brosowski, A., Thrän, D., Mantau, U., Mahro, B., Erdmann, G., Adler, P., Stinner, W., Reinhold, G., Hering, T., and Blanke, C.: A review of biomass potential and current utilisation – Status quo for 93 biogenic wastes and residues in Germany, Biomass and Bioenergy, 95, 257–272, https://doi.org/10.1016/j.biombioe.2016.10.017, 2016.

Bundesministerium für Ernährung und Landwirtschaft: Verordnung über die Anwendung von Düngemitteln, Bodenhilfsstoffen, Kultursubstraten und Pflanzenhilfsmitteln nach den Grundsätzen der guten fachlichen Praxis beim Düngen: Düngeverordnung - DüV, 2017.

Iglesias, J., Martínez-Salazar, I., Maireles-Torres, P., Martin Alonso, D., Mariscal, R., and López Granados, M.: Advances in catalytic routes for the production of carboxylic acids from biomass: a step forward for sustainable polymers, Chemical Society reviews, https://doi.org/10.1039/D0CS00177E, 2020.

Karras, T., Brosowski, A., and Thrän, D.: A Review on Supply Costs and Prices of Residual Biomass in Techno-Economic Models for Europe, Sustainability, 14, 7473, https://doi.org/10.3390/su14127473, 2022.

Scarlat, N., Fahl, F., Lugato, E., Monforti-Ferrario, F., and Dallemand, J. F.: Integrated and spatially explicit assessment of sustainable crop residues potential in Europe, Biomass and Bioenergy, 122, 257–269, https://doi.org/10.1016/j.biombioe.2019.01.021, 2019.

Siegfried, K., Günther, S., Mengato, S., Riedel, F., and Thrän, D.: Boosting Biowaste Valorisation—Do We Need an Accelerated Regional Implementation of the European Law for End-of-Waste?, Sustainability, 15, 13147, https://doi.org/10.3390/su151713147, 2023.

---

## Author Response (AR2)

**Authors response and changes in the manuscript to RC #1 from 24 September 2023**

*1. You mentioned: "The scope of CAFIPLA has been limited to the biomasses mentioned because of their technical and legal suitability for the technologies developed in the project. However, the types of biomass included in the study are determined by the context of the CAFIPLA project. Straw and municipal solid waste are two of the top 10 biogenic residues, by-products and wastes in Europe in terms of their technical potential, according to a literature review by Karras et al. (2022). Outside the context of the CAFIPLA project, it will be useful to extend our model to other biogenic residues, by-products or wastes, such as forest residues or manure, as these biomasses have great potential in Europe. For the units, we believe that the conversion increases the error, as factors, thresholds and restrictions also vary according to the authors' choices. A more detailed explanation is added in line 120ff."*

*• Apparently, audiences won't care or know your project limit. I recommend adding limitations/future work to detail related concerns. The very first question I had was how large the residual biomass potential of the sources studied here is relative to the total potential. I would appreciate a clarification or some discussion of future work on this.*

*• More importantly, the unit conversion uncertainty point is taken, but it is not the reason for not trying. Authors need to understand fresh tons are different units by source and the aggregation of which might not be useful.*

Answer:

We have included information in the manuscript about the limitations and future work of the dataset produced and, in particular, the unit FM. The choice of unit required for further work with our data product is left to the user, as is the choice of conversion factors used. For this study, the focus has been on regionalisation and validation. However, we will consider including more units in future published datasets.

In addition, the work of Karras et al. (2022) has been included in the manuscript, demonstrating the importance of the biomass potential of straw and municipal solid waste in Europe. We agree that consideration of other biomasses such as woody residues and animal by-products should be considered in the expansion of the database.

As discussed in the paper, there are many data gaps, different methodologies, changing definitions of biomasses, and missing or aggregated biomasses in other studies, that make a statement about the total biomass potential questionable, and an estimate of the portion of the biomass potential covered by this study does not, in the authors' view, add value.

In the script several sentences on this topic have been added in the introduction, line 95f and in the discussion 437ff.

*2. Another point related to future work is to scale the data to a global scale. Will that be possible? E.g., FAOSTAT might have some useful source data.*

Answer:

We believe that with current improvements in satellite imagery and classification, forest and agricultural residues can be better estimated and regionalised globally, including FAOSTAT data. However, changes in administrative units must always be taken into account when constructing timeseries. Regionalisation of livestock data from FAOSTAT will already be more difficult on a global scale due to spatial data gaps. For biogenic industrial residues, data problems also remain in FAOSTAT. Biogenic municipal waste has, in our view, one of the greatest future potentials, but the collection rate is very regionally specific, as shown for Europe, and a global estimate will have a high estimation error. Nevertheless, future work in this area is of great interest to us and we will continue to work on it, taking into account your suggestions.

In the script 3 sentences have been added in line 445ff to include this question.